# Hydrogen Fuel Cell Legal Framework in the United States, Germany, and South Korea—A Model for a Regulation in Malaysia

Muhammad Asyraf Azni and Rasyikah Md Khalid *

Faculty of Law, Universiti Kebangsaan Malaysia, Bangi 43600, Malaysia; asyrafazni@gmail.com
* Correspondence: rasyikah@ukm.edu.my; Tel.: +603-89216353

**Abstract:** As a party to the United Nation Framework Convention on Climate Change (UNFCCC), Malaysia is committed to reduce its greenhouse gases (GHG) emission intensity of gross domestic product (GDP) by 45% by 2030 relative to the emission intensity of GDP in 2005. One of the ways for Malaysia to reduce its GHG emission is to diversify its energy mix and to include hydrogen fuel cell (HFC) in its energy mix. Since Malaysia does not have any legal framework for HFCs, it is best to see how other countries are doing and how can it be replicated in Malaysia. This paper reviews the HFC legal framework in the United States, Germany and South Korea as these countries are among those that have advanced themselves in this technology. The researchers conducted a library-based research and obtained the related materials from online databases and public domain. Based on the reviews, the researchers find that these countries have a proper legal framework in place for HFC. With these legal frameworks, funds will be available to support research and development, as well as demonstration of HFC. Thus, it is recommended that Malaysia to have a proper HFC legal framework in place in order to support the development of the HFC industry.

**Keywords:** fuel cell; hydrogen; law; policy; SDG 7; SDG 13

## 1. Introduction

Malaysia ratified the United Nation Framework Convention on Climate Change (UNFCCC) in its mission to reduce the greenhouse gas (GHG) emission. In 2002, the Kyoto Protocol to the UNFCCC emphasized the principle of common but differentiated responsibilities in reducing carbon emission. Malaysia ratified this protocol in 2002 and the Paris Agreement as a commitment to ensure the increase of the world's temperature below 2 degrees Celsius. Under the Paris Agreement, Malaysia is committed to reduce 45% of GHG by 2030 in relation to its 2005 gross domestic product (GDP). In line with this international obligation, Malaysia has taken various steps to fulfill its commitment.

In Malaysia, the energy sector is the biggest carbon dioxide ($CO_2$) emitter since 1990. In 2017 alone, 107 million metric ton (MT) of $CO_2$ was emitted, while the transportation sector emitted 61 MT $CO_2$ [1]. Reducing the $CO_2$ emission will dramatically reduce the GHG emission. Diversifying Malaysia energy generation mix can assist in reducing $CO_2$ emission. The Renewable Energy Act 2011 (the Renewable Energy Act) which encourages the production of renewable energy through feed-in tariff (FIT) scheme. However, the Renewable Energy Act only covers five types of renewable energy which are biogas, biomass, small hydropower, solar photo voltaic, and geothermal energy [2].

In the Ninth Malaysia Plan which ran from 2006–2010, Malaysia was set to achieve 5% of renewable energy in its energy generation mix. In 2010, Malaysia introduced the National Renewable Energy Policy and Action Plan (the Renewable Energy Policy) where it targets 9% of renewable energy in the energy mix by 2020 [3]. To date, Malaysia's renewable energy penetration level is only 2% [4]. It can be concluded that after a decade, Malaysia's share of renewable energy has failed to increase significantly. This indicates

that the Renewable Energy Act and the Renewable Energy Policy has failed to achieve its objective.

Despite this, Malaysia remains committed to increasing the share of renewable energy in the energy generation mix by renewing its target of 20% renewable energy in the energy mix by 2025 [5]. With this renewed commitments, Malaysia should explore other energy options to increase the renewable energy share. One option is to include HFC as seen in several countries including the United States, Germany, and South Korea. In Malaysia, despite the existence of a Fuel Cell Institute at the Universiti Kebangsaan Malaysia, together with huge investment for the HFC development, Malaysia does not have any legal framework to cater for the industry.

Without proper law and policy, private entities would be cautious in making investment, hence impede the development of the HFC industry. In addition, the government also has no proper channel to fund the HFC industry. This paper reviews the HFC legal framework in the United States (US) (and the state of California due to its advancement in the HFC industry), Germany, and South Korea. With these reviews, the researchers hope to recommend a proper legal framework to support the development of HFC industry in Malaysia.

## 2. Methodological Approach and Used Materials

This paper employs a library-based research where the researchers find all the materials needed from primary sources (such as statutes, regulations, and policies) and secondary sources (such as online databases). The researchers have relied on online databases such as ScienceDirect [6], Scopus [7], and Web of Science [8] to search for relevant materials. Among the keywords use for the searches are 'United States hydrogen', 'United States fuel cell', 'Germany hydrogen', 'Germany fuel cell', 'South Korea hydrogen', and 'South Korea fuel cell'. From these materials gathered, the researchers then used the ATLAS.ti software to code them into specific codes such as 'energy body', 'energy regulatory framework', 'hydrogen', 'hydrogen body', 'overview', 'RE goals' and 'RE status'. These materials are then being grouped based on their countries into code groups such as 'US', 'California', Germany', and 'Korea'. By grouping them in accordance with their country (and state as for California), this helps the researchers to extract relevant information to be used in the result's part of this paper. This paper does not follow the strict methods of doing reviews in social sciences' discipline. As the researchers are from legal background, the researchers use a legal methodology which is a doctrinal method. Although this paper is a qualitative in nature because there is an involvement of reading materials (rather than field work such as survey in quantitative method); however, it is also a doctrinal in nature because it involves the reading and interpreting laws, acts, or provision of laws. These research methods are use in order to answer two research problems of this paper. Firstly, how do the United States, Germany, and South Korea support the development of HFC industry in their country? Secondly, what are the advantages obtained by each country by having a proper legal framework on HFC? With regards the selection of countries in this review, there are four criteria that the researchers consider. Firstly, the countries have policies or laws on the HFC industry. Secondly, the materials to be reviewed are available in English. Thirdly, the countries (at least one country) must have the same system of government as Malaysia. Fourthly, the countries are from different regions of the world. As for the first criteria, the US, Germany, and South Korea are all have policies and/or laws related to HFC industry. As for the second criteria, the materials in English are easily accessible by the researchers for the three countries stated above. Although Japan is one of the leading countries in HFC industry; however, there are limited materials in English. Thus, there is not much that can be analyzed for the purpose of this review. As for the third criteria, the US has the same system of government as Malaysia, which is federalism. Last but not least, the US, Germany, and South Korea are from different continents which are the North America, Europe, and Asia, respectively.

### 3. An Overview of Hydrogen Fuel Cell Legal Frameworks in the US, Germany, and South Korea

*3.1. The Legal Framework on Hydrogen Fuel Cell in the US*

3.1.1. The Federal Legal Framework on Hydrogen Fuel Cell in the US

The US has started to explore the HFC industry since 1990 as it realizes the benefit of hydrogen. This is evident by the enactment of the Spark M. Matsunaga Research, Development, and Demonstration Program Act by the federal government. The name of the Act was taken from the name of Senator Spark M. Matsunaga who promoted hydrogen to replace fossil fuels and reduce its side effects [9]. In 1992, the federal government enacted the Energy Policy Act to support the development of HFC. In 1996, the Hydrogen Future Act was enacted, which among others, direct the Secretary of Energy to conduct a research, development, and demonstration (RD&D) program on the production, storage, transportation of hydrogen, and the use of hydrogen for industrial, residential, transportation, and utility application sectors [10].

In 2001, the National Energy Policy was established pursuant to the President's aspiration to have an energy policy that plans for the future and fulfils the present needs [11]. In February 2002, the federal government, through the Department of Energy, produced the 'National Vision of America's Transition to a Hydrogen Economy—to 2030 and beyond' that addressed two main problems; reliance on petroleum import, and GHG emission and pollution. This plan suggests, among others, for the establishment of national hydrogen energy roadmap to continue the transition toward hydrogen [12]. Based on this vision, the National Hydrogen Energy Roadmap was produced in November 2002 to encourage all stakeholders to work together to reduce risk, improve performance, reduce costs and implement a reliable, clean, and trusted energy of the future [13].

The same year, the Freedom Cooperative Automotive Research (FreedomCAR) under the partnership between the Department of Energy and the US Council for Automotive Research. FreedomCAR enables the transition to hydrogen economy, and ensuring easy access to HFC. In 2003, FreedomCAR expanded their work with the oil industries such as BP America, Chevron, ConocoPhillips, Exxon Mobil, and Shell Hydrogen LLC. Together they develop the component and infrastructure's technology needed to reduce the US reliance on oil import and reduce GHG emission from vehicles, without sacrificing the freedom of mobility and choosing a vehicle among the citizen [14]. President George W. Bush also announced the Hydrogen Fuel Initiative worth USD 1.2 billion to enhance the HFC technology for cars, trucks, homes, and businesses [15].

In 2005, the federal government enacted another law named Energy Policy Act [16] to fully support the HFC development. Section 731 (a) directs the Secretary to establish demonstration program of 25 fuel cell buses. Section 731 (c) provides USD 10 million funding for the program until 2010. As of 2018, there were 32 fuel cell buses operating in the US [17]. Section 741 further defines hydrogen as an alternative fuel. This is essential to ensure that hydrogen will receive necessary benefit provided under this Act. Next, Section 802 provides for the promotion of the development of the HFC industry with the industry players, while Section 804 directs the Secretary to submit a coordinated plan to the Congress on the HFC programs. Pursuant to Section 805 (a), USD 160 million were provided in 2006, and the sum has increased to USD 200 million in 2007, USD 220 million in 2008, USD 230 million in 2009, and USD 250 million in 2010. From 2011 until 2020, the fund will be available upon request. These legal and financial mechanisms ensure that the success of the Act.

Section 806 of the Act also establishes a Hydrogen and Fuel Cell Technical Task Force, which is, among others, responsible to develop a safe, economy, and environmental-friendly infrastructure for the HFC industry. Furthermore, Section 807 (a) created a Hydrogen Technical and Fuel Cell Advisory Committee to advise the Secretary of Energy on programs and activities under the Act. Under Section 808 (a), the Secretary is required to finance certain demonstration projects consistent with the Act. Consequently, USD 185 million were provided in 2006, USD 200 million in 2007, USD 250 million in 2008, USD 300 million

in 2009, and USD 375 million in 2010. From 2011 until 2020, the fund will be available upon request.

Concerning the code and standards for the HFC industry, Section 809 allows the Secretary to provide a grant or contract with the professional organizations, public sector, and government agencies to support the development of a comprehensive codes and standards for fuel cell electric vehicle (FCEV), hydrogen energy system and stationary, mobile, and micro fuel cell. Under Section 809 (c), USD 4 million were passed in 2006, USD 7 million in 2007, USD 8 million in 2008, USD 10 million in 2009, and USD 9 million in 2010. For 2010 until 2020, the amount will be available upon request. Section 811 (a) requires the Secretary to submit a report to the Congress every 2 years on the activities carried out by the Department of Energy under this Act. This ensures effective monitoring mechanism and successful execution of all programs.

In 2006, President Bush produced the Advanced Energy Initiative where he increased the fund to 22% for funding RD&D on clean energy technology through the Department of Energy. This initiative also aims at reducing fossil fuel-powered vehicles to hydrogen-powered vehicles by 2020 [18]. In 2011, the Hydrogen and Fuel Cells Program Plan was established to ensure a more strategic and integrated plan for the RD&D of the HFC industry [19]. Since 2016, the H2@Scale program was implemented to bring together various stakeholder to fully develop the HFC industry from the production, storage, transportation, and application [20]. Recently, in July 2020, approximately USD 64 million funding were given to 18 projects under the H2@Scale on affordable hydrogen production, storage, transportation, and application [21]. The Fuel Cell and Hydrogen Energy Association, group of private companies and organizations that produce innovative, clean, safe, and reliable energy, has produced a Road Map to a US Hydrogen Economy to achieve US mission of HFC economy [22].

### 3.1.2. The Legal Framework on Hydrogen Fuel Cell in the State of California

Based on research, California has one of the most advanced legal frameworks for HFC. California used to have a pollution problem for years and has taken strong action to clean its air. The journey toward hydrogen begins as one of its effort in emission reduction. In 1999, the California Fuel Cell Partnership was established between the government agencies, academicians, industry players, and non-governmental organization [23]. To increase the Zero Emission Vehicle (ZEV), the Californian Hydrogen Highway Network program was introduced in 2004 [24] and developed hydrogen refueling stations on the chain of highways in California. This allows easy access to the hydrogen refueling stations and faster transition toward complete FCEV.

In 2005, the Hydrogen Highway Blueprint Plan suggested the following:

i.   sharing of cost for the building of hydrogen refueling stations between the public and private sector;
ii.  USD 10,000 incentive for each car for California to achieve 2000 Fuel Cell Electric Vehicle (FCEV) within five years; new laws and policies that attract businesses to join the HFC industry;
iii. setting up an insurance fund for the owners of hydrogen refueling station; and;
iv.  more outreach program for greater public participation in the transition to HFC economy [25].

In 2006, California introduced the Senate Bill 1505 [26] to encourage the production of hydrogen from the renewable sources based on the Blueprint. Thus, at least 33% of hydrogen produced in California must come from renewable sources [27]. In March 2012, the Californian Governor Brown Jr. issued an executive order which among others, provided for California to have 1.5 million ZEV by 2025 [28]. Later, the California Road Map: Bringing Hydrogen Fuel Cell Electric Vehicles to the Golden State was produced [29]. In 2013, Assembly Bill 8 was accepted into law which among others, provides USD 20 million a year until 100 hydrogen refueling stations were built in California [30].

In 2018, California Fuel Cell Partnership produced The California Fuel Cell Revolution: A Vision for Advancing Economic, Social and Environmental Priorities [31]. In its 2030 vision, 1000 hydrogen refueling stations will be developed to serve 1 million FCEV. In the same year, the Energy Independence Now, in collaboration with the Leonardo DiCaprio Foundation and the California Hydrogen Business Council, produced the Renewable Hydrogen Roadmap, to promote the renewable hydrogen production [32]. These initiatives are so important in realizing FCEV in the state.

### 3.2. The Legal Framework on Hydrogen Fuel Cell in Germany

Germany is among the most advanced countries in HFC as it has already embarked on the HFC economy. In 2006, the German government, together with industry players and stakeholders, formed an alliance and established the National Innovation Program Hydrogen and Fuel Cell Technology (National Innovation Program) [33]. The objectives of the National Innovation Program are (1) to put the Germany in the HFC industry as a technology leader, (2) to speed up the HFC market development, and (3) strengthening the HFC industry [34].

In 2008, the National Organization Hydrogen and Fuel Cell Technology, or locally known as *Nationale Organisation Wasserstoff- und Brennstoffzellentechnologies* (NOW), was established, wherein its sole shareholder is the Federal Ministry of Transport and Digital Infrastructure (*Bundesministerium für Verkehr und digitale Infrastruktur* (BMVI)). NOW is responsible for designing, coordinating, and executing the national strategies and programs between the public and private sector in the field of technology for sustainable mobility and energy supply [35,36]. NOW also helps the BMVI and the Federal Ministry for Economic Affairs and Energy (*Bundesministerium für Wirtschaft und Energie* (BMWi)) in implementing the National Innovation Program. The BMWi focuses on applied research and development, while the BMVI is focusing on projects demonstration and development of the market [34].

In September 2010, Germany produced a document called the Energy Concept to secure an energy source that is affordable, reliable, and environmental-friendly. Germany's electric mobility strategy aims to secure one million electric vehicles by 2020 and six million by 2030. Germany believes that FCEV can be the vehicle of the future as long as it is powered by the renewable hydrogen [37]. Furthermore, in 2016, Germany produced the Climate Action Plan 2050 to reduce emission from transportation comprises of 30% of the energy use and 90% of it are based on petroleum [38].

Under the Electric Mobility Act of 2015 [39], vehicles are labeled with 'E' which was taken from the term of 'electric mobility' or 'eMobility'. Since 2009, the government has provided EUR 5 billion and various legal frameworks to support the electric mobility and entice the public. From 2009 until 2020, EUR 300 million are allocated to develop electric vehicle charging infrastructure. The government has provided incentive for purchasing plug-in hybrid and fully electric car, and the cost is shared equally between the government and the automotive industry. The cost is EUR 4,000 for fully electric car (powered by batteries or fuel cell) and EUR 3,000 for plug-in hybrid [40].

According to the report coordinated by NOW on the evaluation of National Innovation Program from 2006–2016, since 2006 until 2016, the BMVI and BMWi has funded almost EUR 710 million to about 750 research and development (R&D) projects. The recipients of this grant have invested EUR 690 million and received EUR 20 million via the third-party funding. The National Innovation Program comprises of various application areas such as hydrogen production, application in transportation sector, building heat and power application, industry heat and power application, and specialized markets. This program made Germany as the 3rd largest country to fund HFC technology, behind Japan and the US [34].

To lead by example and encouraging the market, the government has invested EUR 100 million for its vehicles. Germany aims for 20% of its vehicles are from electric vehicle (plug-in hybrid and FCEV) with 74% shares in the total vehicles belonging to Federal Ministry of Transportation [40]. The government has also produced the BMVI's Charging

Infrastructure Funding Guidelines to increase the number of charging station for the current and future needs. Until 2020, EUR 300 million is allocated to develop accessible 15,000 charging station [40]. Recently, in June 2020, the government has launched the National Hydrogen Strategy which provided EUR 9 billion with the focus of producing renewable hydrogen [41].

*3.3. The Legal Framework on Hydrogen Fuel Cell in South Korea*

Since 1973, South Korea has considered the HFC as an engine for the growth of new economy and alternative energy sources. This is evident by the fact that since 2003 until 2012, the government has allocated R&D fund for the HFC. Through the Ministry of Science and Technology, a Hydrogen R&D Center has been established at the Korea Institute of Energy Research in 2003 [42]. This institute is the government-funded national research institute for hydrogen energy technology and develop policies for the government. The Fuel Cell R&D Center was also established [42,43], and the government has invested USD 100 million for Hydrogen R&D Center between 2003 and 2012 [44].

The law that governs the HFC market is the Alternative Energy Technology Development and Usage Promotion Act 1987 [45,46]. Unlike other countries that uses the term 'renewable energy', South Korea uses the term 'new and renewable energy'. 'New energy' here is referring to among others, hydrogen energy and fuel cell. Meanwhile, 'renewable energy' comprises of solar, wind, hydro, ocean, geothermal, bio energy, and energy from waste products [47,48].

In 2003, the second Basic New and Renewable Energy Plan is published to support the development of NRE in South Korea's market. Since the change in policy, the investment in HFC R&D has increased with USD 110.8 million invested in 2007 alone. In 2005, the Hydrogen Economy National Vision and Execution Plan [46] was produced to increase understanding of the public on the HFC economy and to strategize the HFC development [43].

In January 2019, the government has produced the Hydrogen Economy Roadmap 2040 of Korea (the Roadmap) [38]. The main objective of the Roadmap is to establish a hydrogen industry ecosystem comprises of production, storage, transport, safety, and to maximize the hydrogen benefit which is its mobility [49]. The same year, the amount of subsidy given by federal and local government for the purchase of FCEV is around USD 27,300—USD 30,300, almost half from the sale value of Hyundai NEXO (around USD 59,000). Due to this, the annual sales of NEXO has reached 4987 units in 2019 surpassing Toyota MIRAI at 2,494 units [38]. Other news outlet reported that between January until October 2019, South Korea has recorded the sale of 3666 unit of hydrogen cars which is equivalent to 60% of global sales and has overtaken Japan in term of global sales [35,50].

The Roadmap runs with the third Basic Energy Plan (2019–2040) which put hydrogen in its future energy mix. Basic Energy Plan is the plan that shape the whole energy system in South Korea. Unlike the Basic New and Renewable Energy Plan that focuses on the development of new and renewable energy, Basic Energy Plan's coverage and scope is much wider. The government has also produced the National Roadmap of Hydrogen Technology Development [38] to develop a Hydrogen Model City in Ulsan, Alsan, and Wanju-Jeonju area by 2020 [51].

In 2019, 20 hydrogen refueling stations were built, which is the highest number of hydrogen refueling stations built in a year in the world. This makes the total hydrogen refueling stations operated in South Korea 34 units [35,50]. Half of the cost of hydrogen refueling stations is subsidized by the government. In March 2019, a specialized-aim company, Hydrogen Energy Network, was established with 13 companies to construct around 100 hydrogen refueling stations by 2022 [38].

In January 2020, the government enacted the Hydrogen Economy Promotion and Hydrogen Safety Management Act (the Hydrogen Economy Act) [52] that enables the government to provide support for the HFC development and facilities for safety and standard. [38]. This however will only come into force on 5 February 2021 to give ample

time for the relevant ministries to put proper regulations in place. The act will establish the Hydrogen Economy Council for the purpose of planning strategies, policies, and regulations. This act also supports the HFC-specialized companies through subsidies, loans, and tax exemptions. Furthermore, any company who wants to invest in the HFC-specialized company can be registered and designated as the hydrogen investment company. The act also covers the safety standards in areas such as hydrogen-related business permits, various hydrogen-related facilities, and inspections at those facilities. Failure to comply with its standards will be penalized accordingly and the requirement of mandatory insurance coverage for these facilities. [53].

## 4. Discussion

### 4.1. Legal Framework for Advancing the Hydrogen Fuel Cell Industry in the US

It is acknowledged that the US has been exploring into hydrogen as early as 1990. The Spark M. Matsunaga Research, Development, and Demonstration Program Act and Energy Policy Act of 1992 were enacted to support the early development of HFC industry. It is evident that the existence of these laws help to support the RD&D in HFC industry. The coverage of the laws is wide as they cover the aspect of production, storage, and transportation of hydrogen.

The HFC industry in the US receives continuous support from the government with the introduction of Energy Policy Act of 2005. Millions of dollars were allocated for HFC development and various bodies were established to support the industry, including the Hydrogen and Fuel Cell Technical Task Force and the Hydrogen Technical and Fuel Cell Advisory Committee. These bodies would not have come into existence should there is no law enacted as their source of power.

Besides these legal documents, there are also other legal frameworks supporting the HFC industry. This can be seen from the existences of various policies and plans such as the 'National Vision of America's Transition to a Hydrogen Economy—to 2030 and beyond', National Hydrogen Energy Roadmap, Hydrogen Fuel Initiative, Advanced Energy Initiative and the Department of Energy Hydrogen and Fuel Cells Program Plan. The difference between these policies and plans with the Energy Policy Act of 2005 is that the Act is a general law that caters the whole energy aspect, while the policies and plans are specifically focusing on the HFC industry.

When it comes to effectiveness, the laws have more effect in ensuring the success rate of any plan. However, like many countries, the government is not willing to devote themselves fully to the HFC industry. The researchers are of the opinion that perhaps they are not 100% confident that hydrogen is a way forward. Besides, financial allocation is also needed in other sectors. Policies and plans do not really bind the government to fulfill what has been targeted. Nonetheless, they provide good guidance for future needs and funding. In all, it is safe to conclude that by with legal frameworks be it laws, policies, or plans, it enables the US to provide support in the development of HFC industry.

The existence of laws, policies, and plans from the government have also increased the level of confidence among the private sector to invest in the HFC industry. Established companies such as BP America, Chevron, ConocoPhillips, Exxon Mobil, and Shell Hydrogen LLC have formed a partnership with the government to work together in realizing the establishment of a hydrogen economy. The private sector and various stakeholders have also supported the HFC industry when the Fuel Cell and Hydrogen Energy Association produced a Road Map to a US Hydrogen Economy. This shows that they are serious in moving toward hydrogen economy. Based on the recent allocation of funding, approximately USD 64 million, to 18 hydrogen-related projects by the Department of Energy, the HFC sector in the US is still alive and kicking despite the worries by certain quarters that the President would not support the development HFC industry. It is hoped that the US government would continue its support for the HFC industry.

The same result can be seen in California. Determined to protect the local environment, California established the Low Carbon Fuel Standard. This brought California ahead of

other States and the federal government. It aims to reduce the GHG emission by mandating automakers to produce cars which emit low carbon. As time goes by, the law has become stricter in mandating the automakers in producing the ZEV. The law also puts a lower GHG emission rate target as compared to the rate targeted by the federal government. Although the law is not specifically focusing on HFC (because it aims is to reduce the GHG emission); however, it allows California to actively supporting the HFC industry as a way to reduce the GHG emission.

California has also introduced various policies, plans, and programs to support the development of HFC industry. On top of that, the law such as Senate Bill 1505 and the executive order by the Californian Governor Brown Jr. and Assembly Bill 8 also put in place sufficient support for the HFC industry. With these laws and other legal frameworks, the government of California has funded massively on the HFC industry. In return, the private sectors supported the government efforts in moving toward hydrogen economy. This can be seen by the establishment of the California Fuel Cell Partnership, which has been so active in HFC industry in California.

### 4.2. Legal Framework for Advancing the Hydrogen Fuel Cell Industry in Germany

Moving on to Germany, the HFC has been supported through their national policy named Energy Concept. Although this is just a policy and does not have a legal effect as compared to any laws, it provides a new energy pathway in Germany. It guides the government on prioritizing things to achieve energy security and at the same time environmental-friendly. Based on the result stated in Section 3.2, this policy has identified FCEV as the vehicle for the future provided that the hydrogen is produced from renewable sources.

Germany also supports the HFC industry with the enactment of Electric Mobility Act of 2015 and provides subsidies for the purchase of FCEV. The law allows the government to provide billions of euros to move toward electric vehicles. From here, it shows that the government has supported the development of HFC industry via the establishment of policy and law. All these efforts justify Germany's action in an effort to reduce the GHG emission.

In return, the private sector in Germany is more willing to invest and take part in the HFC industry. This can be seen from the establishment of the National Innovation Program. This program has made Germany as the third largest country in the world in providing funding for the HFC technology. The latest development from Germany also shows that it is serious in developing the HFC industry with the establishment of the National Hydrogen Strategy in June 2020. Although this is also a policy, and not a law, it allows the government to provide EUR 9 billion for the development of renewable hydrogen.

### 4.3. Legal Framework for Advancing the Hydrogen Fuel Cell Industry in South Korea

Last but not least, South Korea. Based on Section 3.3 of this paper, it can be seen that the South Korea has been supporting the HFC industry by having various laws and policies in place and this can be seen since as early as 1973. The HFC industry is further supported by the establishment of centers such as Hydrogen R&D Center and Fuel Cell R&D Center, focusing on the development of HFC. The existence of policies such as Second Basic New and Renewable Energy Plan have enabled the government to provide millions of dollars to support the development of HFC industry.

The usage of the term 'new and renewable energy' in South Korea also shows that the alternative sector in Korea is much wider than in most countries that focused on renewable energy only, including Malaysia. With this wider term, South Korea has the advantage of diversifying its alternative sources to ensure that it achieves its energy security and fulfill its international obligation in reducing the GHG emission.

Furthermore, South Korea can be seen as very serious in moving toward a hydrogen economy when it put in place the Hydrogen Economy Roadmap 2040 in 2019 followed by the enactment of Hydrogen Economy Promotion and Hydrogen Safety Management Act

in January 2020. Compared to the US and Germany, which do not have specific laws on the HFC, South Korea has moved one step ahead by enacting, in details, various aspect of HFC. However, as the law can only be accessible in Korean language, the researchers can only rely on the help of Google Translate to examine the law. Nonetheless, based on the translated version of the law, this law allows the government to establish a special body on HFC, it also allows the government to support the development of HFC and HFC-specialized company and provide an allocation for the education program on HFC. The law also establishes the safety center in order to ensure the safety aspect of HFC are being prioritized. South Korea has once again shown that it is determined to establish HFC economy through this Act and to spur the development of HFC in South Korea.

*4.4. Current Market for Hydrogen FCEV*

The government of US, Germany, and South Korea are all putting the focus on transforming the transportation industry from diesel-engine vehicle into FCEV. There are two factors to this: the transport sector is the biggest $CO_2$ contributor in most cities, and transportation is indispensable for the general public. Compared to the HFC-based power plant, vehicles are closer to the general public. Thus, if the governments can convince the public to accept this HFC technology, it would be easier to spur the development of HFC industry. In reality, lack of public support has impeded many government's plan.

As for the status of FCEV in the US federal government and/or the Californian state government, up until 1 October 2020, there are currently 8654 fuel cell cars that have been sold or leased in the US and 48 fuel cell buses operating in California. In California alone, there are 42 hydrogen refueling stations (HRS) are operating [54].

Moving on to Germany, as of June 2019, there are 386 registered FCEV in Germany [55]. As for HRS, up until 2019, Germany has the second largest HRS available in the world with 60 HRS operating, just behind Japan with 96 HRS. Out of this, there are 17 HRS come into operation in 2018 alone [56].

Last but not least, for South Korea, as in 2019, the total FCEV sold in South Korea has reached more than 5000 units as compared to only 87 units in 2016 [57]. In terms of HRS, South Korea has 39 station operating where 20 stations were built in 2019 alone.

Taking stock of number of FCEV in these countries, it is argued that specific laws policies on HFC can assist in advancing the HFC industries. Compared to other countries that do not have any legal framework on HFC, it is unlikely that FCEV will run on their highways. The ability of the car manufacturers to produce FCEV in particular, and HFC technology in general, is very much depend on a clear government framework for this industry, Li (2021) argues that HFC technology today has matured and can move to mass production and commercialization. What is lacking is the commitment and political will in transitioning into the hydrogen economy [58]. If countries continue to invest on the heavily subsidized fossil fuel-based industry, the HFC industry could never compete on a fair ground. In addition, if governments continue to shape the policies and laws as lobbied by the fossil fuel-based industry, there shall be no real support from the government for the development of the HFC industry.

Similarly, FCEV will not be able to compete with the battery electric vehicles (BEV) without the same support from the government and public awareness on FCEV and HFC. As of 2019, there are five million BEV sold in the world if compared to 7500 FCEV sold [59]. The reasons for this blatant difference is obvious; the BEV is more popular with the general public and they are more familiar with the range, price, performance and re-charging facility of BEV. Contrary to the public perception, FCEV prevails over BEV in term of performance and recharging or refueling conveniences. Nevertheless, BEV has greater range and the cost of owning and maintaining BEV are much cheaper than FCEV [60].

With regard to the range, there is an urgent need to educate the public about HFC and FCEV. When the public knows about the advantages of HFC technology, the public will be more willing to accept the technology. This can be evident in the case of BEV. With regard to the refueling facility, the FCEV is not attractive because of limited HRS. This is a

"chicken and egg" problem that could be solved by investing more on HRS. Experts and car manufacturers have predicted that the price of FCEV will go down as the production increases. Thus, it is crucial that HFC technology is being mooted out as the energy of the future because of the world's fight against climate change. Both FCEV and BEV will work together to achieve sustainable development goals (SDG) 7 on clean energy and SDG 13 on climate action. Mr. Mark Kirby, the President and CEO of Canadian Hydrogen and Fuel Cell Association has rightly put it that this is not a fight between FCEV and BEV, but this is a fight between FCEV and BEV against fossil fuels [58].

### 4.5. The Status of HFC Industry in Malaysia

Malaysia is one of rapidly developed Southeast Asia countries. The relocation of the administrative capital from Kuala Lumpur to Putrajaya, and the development of the new Kuala Lumpur International Airport in Sepang, have expanded the conurbation, but also bring new problems including water and air pollution [61]. Rapid increase in transportation, which is the biggest $CO_2$ emitter, is inevitable. There has been initiative towards low carbon cities through a real time carbon abatement measures, but the progress is very slow [62]. Since HFC and FCEV are carbon free, they are regarded as the best mechanism to speed up emission reduction.

In 2010, Malaysia introduced the Renewable Energy Policy and enacted the Renewable Energy Act in 2011. After a decade, Malaysia's share of renewable energy in the energy generation mix has not increase as hoped. The Renewable Energy Act is also limited in increasing five types of renewable energy which are biogas, biomass, hydropower, solar photo voltaic, and geothermal energy. It does not facilitate the use of alternative and carbon-free energy such as HFC.

The emergence of HFC can be seen as early as 1995 when Malaysia built its first sin-gle cell proton exchange membrane fuel cell and developed the advanced materials for polymer composite bipolar palates. This started upon research collaboration between the Universiti Kebangsaan Malaysia and the Universiti Teknologi Malaysia funded by the Ministry of Science, Technology and Environment (MOSTI) under the Seventh Malaysia Plan [63]. The Malaysia Plan is a five-year development plan that highlights the govern-ment's budget commitment for every five years for the economy, society and environment including the water and energy sector [64]. HFC was recognized as a 'potential alternative energy' in the Eighth Malaysia Plan (2001-2005). In the following Ninth Malaysia Plan (2006-2010), the government introduced financing mechanism for the technology development and sharing of knowledge of HFC [65].

Under the Tenth Malaysia Plan (2011–2015) and Eleventh Malaysia Plan (2016–2020), HFC was no longer mentioned. Nevertheless, FCEV was developed as a demonstration project by the national automaker, PROTON. Although HFC was no longer highlighted in the development plan, MOSTI has recognized HFC as one of its priority research areas and has allocated RM41 million funding from 1995 to 2017 [65]. In 2017, the Academy of Sciences Malaysia (ASM) published the Blueprint for Fuel Cell Industries in Malaysia. Although it is not a national plan for the HFC industry, it lays down the short, medium and long-term goals for the development of HFC industry in Malaysia until 2050 [66].

Malaysia's sluggish progress in achieving its renewable energy target and the absence of HFC in the recent government's development plan could be due to the fact that Malaysia is rich in natural gas and crude oil. In 2011, Malaysia ranked 28th in the world and 4th in the Asia-Pacific after China, Indonesia, and India in crude oil production. For natural gas, Malaysia ranked 11th in the world and 3rd in the Asia-Pacific, behind China and Indonesia [67]. This leads to "resource curse" or "paradox of plenty" that leads to crowding-out effect when billions are spent on fuel subsidies. This hinders investment in the renewable or alternative energy sector [68].

Meanwhile, the government-linked power company continues to lobby for policies favorable to them. This will thwart government efforts to formulate policies or laws to increase the renewable and alternative energy share in the national energy mix. What

is now needed is a strong political commitment to bring the laboratory works on HFC into real life. This can be seen in the state of Sarawak where the Chief Minister has been championing HFC through the state-linked energy producer, Sarawak Energy. There are now several FCEV on the road of Sarawak and this has encouraged stakeholder to take part in the hydrogen industry and economy.

## 5. Conclusions

The US, Germany, and South Korea have been successful in advancing HFC from merely a laboratory exercise to a profitable industry. This is partly contributed to the political will in ensuring better awareness and acceptance towards HFC. This is also contributed by well drafted laws and policies to regulate and incentivize the industry players and consumers. These include the 2001 National Energy Policy and the Energy Policy Act 2005 in the US; the Electric Mobility Act and the Climate Action Plan 2050 in Germany; and the Alternative Energy Technology Development and Usage Promotion Act 1987 and the Hydrogen Economy Promotion and Hydrogen Safety Management Act 2020 in South Korea.

Malaysia is ambitious to reduce its carbon emission, but there is no significant commitments for climate mitigation, for example in the energy and manufacturing sectors, on carbon tax, or innovations in the carbon-saving technology [69]. Malaysia government need to embark on a more serious endeavor in promoting HFC as in the US, Germany, and South Korea. By providing a friendly legislative framework that entices both industries and consumers to invest in the HFC industry, Malaysia will also achieve its Paris Agreement commitment to reduce carbon emission by 45% in 2030. A proper roadmap on the HFC industry, and perhaps the hydrogen economy, can increase the level of confidence among the investors in the sector. This should be followed by a proper law, which are not only regulate, but also facilitate the sector.

Malaysia should also leverage on its tropical weather where it has abundant sunlight that could be used for large solar plant to produce renewable hydrogen as compared to the northern countries. By moving forward,, a decade of research and development on HFC worth millions of dollars will not go in vain. As a matter of fact, researchers in Malaysia has been advancing the fuel cell research beyond HFC to include methanol [70] and Nafion membrane [71] Rapid development of renewable and alternative energy will also assist Malaysia towards achieving SDG7 on clean energy and SDG 13 on climate action.

**Author Contributions:** M.A.A. (methodology, software, formal analysis, writing—original draft preparation) and R.M.K. (conceptualization, writing and editing, project administration, funding acquisition). All authors have read and agreed to the published version of the manuscript.

**Funding:** This research was funded by Ministry of Higher Education of Malaysia and Universiti Kebangsaan Malaysia (grant number: TRGS/1/2018/UKM/01/6/3).

**Institutional Review Board Statement:** Not applicable.

**Informed Consent Statement:** Not applicable.

**Data Availability Statement:** Not applicable.

**Acknowledgments:** The authors would like to thank their family and friends who had supported the authors directly or indirectly in the preparation of this paper.

**Conflicts of Interest:** The authors declare no conflict of interest. The funders had no role in the design of the study; in the collection, analyses, or interpretation of data; in the writing of the manuscript, or in the decision to publish the results.

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
