# Peer review of "Hydrogen Fuel Cell Legal Framework in the United States, Germany, and South Korea—A Model for a Regulation in Malaysia"

_sustainability, doi:10.3390/su13042214_

Round 1
Reviewer 1 Report
This manuscript reviews the legal framework on hydrogen fuel cell in the United States, Germany, and South Korea aiming to advice Malaysia to introduce hydrogen fuel cell in its energy mix.
- Please use full name “Renewable Energy” not abbreviation “RE” considering that the full name is not long.
- Japan is one of the leading countries in the area of hydrogen fuel cell technologies. Why was the legal framework on hydrogen fuel cell in Japan not mentioned in this review?
Author Response
Dear Sir,
Please see attachment.

Reviewer 2 Report
This is a very interesting topic. The paper summarizes the state of programs to subsidize the development of hydrogen vehicles in the US, Germany and South Korea. The paper could be greatly improved if it would examine why these programs have not been very successful in stimulating the development of such technology. The number of hydrogen-fueled vehicles in each country is dwarfed by the number of electric vehicles. As of 2 years ago there were only 7,500 hydrogen vehicles sold in the world compared to 5 million electric vehicles. What explains this? Were the subsidies for hydrogen-fueled vehicles too low? Are there problems with the technology? What is Malaysia doing, if anything, to promote the use of hydrogen?
Author Response
Dear Sir,
Please see attachment.
Thank you for your time.

Reviewer 3 Report
The paper addresses a very topical research question. The paper is well-written and has a usual structure. Nevertheless, the issue of country type has not been addressed sufficiently. Malaysia is a fossil fuel abundant country. To close this gap refer to Sadik-Zada and Gatto (2020) and Sadik-Zada and Lorwenstein (2020) and drop a few words in this regard.
Author Response

(The authors gave the same response as above.)

Round 2
Reviewer 2 Report
This version is significantly improved and should be published.